# In Defense of Pseudo-Labeling:
# An Uncertainty-Aware Pseudo-label Selection Framework for Semi-Supervised Learning

**Mamshad Nayeem Rizve[†], Kevin Duarte[†], Yogesh S Rawat[‡] & Mubarak Shah[‡]**
Center for Research in Computer Vision
University of Central Florida, Orlando, Florida, USA
[†]{nayeemrizve, kevin_duarte}@knights.ucf.edu
[‡]{yogesh, shah}@crcv.ucf.edu

## Abstract

The recent research in semi-supervised learning (SSL) is mostly dominated by *consistency regularization* based methods which achieve strong performance. However, they heavily rely on domain-specific data augmentations, which are not easy to generate for all data modalities. Pseudo-labeling (PL) is a general SSL approach that does not have this constraint but performs relatively poorly in its original formulation. We argue that PL underperforms due to the erroneous high confidence predictions from poorly calibrated models; these predictions generate many incorrect pseudo-labels, leading to noisy training. We propose an uncertainty-aware pseudo-label selection (UPS) framework which improves pseudo labeling accuracy by drastically reducing the amount of noise encountered in the training process. Furthermore, UPS generalizes the pseudo-labeling process, allowing for the creation of negative pseudo-labels; these negative pseudo-labels can be used for multi-label classification as well as negative learning to improve the single-label classification. We achieve strong performance when compared to recent SSL methods on the CIFAR-10 and CIFAR-100 datasets. Also, we demonstrate the versatility of our method on the video dataset UCF-101 and the multi-label dataset Pascal VOC.

## 1 Introduction

The recent extraordinary success of deep learning methods can be mostly attributed to advancements in learning algorithms and the availability of large-scale labeled datasets. However, constructing large labeled datasets for supervised learning tends to be costly and is often infeasible. Several approaches have been proposed to overcome this dependency on huge labeled datasets; these include semi-supervised learning (Berthelot et al., 2019; Tarvainen & Valpola, 2017; Miyato et al., 2018; Lee, 2013), self-supervised learning (Doersch et al., 2015; Noroozi & Favaro, 2016; Chen et al., 2020a), and few-shot learning (Finn et al., 2017; Snell et al., 2017; Vinyals et al., 2016). Semi-supervised learning (SSL) is one of the most dominant approaches for solving this problem, where the goal is to leverage a large unlabeled dataset alongside a small labeled dataset.

One common assumption for SSL is that decision boundaries should lie in low density regions (Chapelle & Zien, 2005). Consistency-regularization based methods achieve this by making the network outputs invariant to small input perturbations (Verma et al., 2019). However, one issue with these methods is that they often rely on a rich set of augmentations, like affine transformations, cutout (DeVries & Taylor, 2017), and color jittering in images, which limits their capability for domains where these augmentations are less effective (e.g. videos and medical images). Pseudo-labeling based methods select unlabeled samples with high confidence as training targets (pseudo-labels); this can be viewed as a form of entropy minimization, which reduces the density of data points at the decision boundaries (Grandvalet & Bengio, 2005; Lee, 2013). One advantage of pseudo-labeling over consistency regularization is that it does not inherently require augmentations and can be generally applied to most domains. However, recent consistency regularization approaches tend to outperform pseudo-labeling on SSL benchmarks. This work is in defense of pseudo-labeling: we demonstrate that pseudo-labeling based methods can perform on par with consistency regularization methods.

Although the selection of unlabeled samples with high confidence predictions moves decision boundaries to low density regions in pseudo-labeling based approaches, many of these selected predictions are incorrect due to the poor calibration of neural networks (Guo et al., 2017). Since, calibration measures the discrepancy between the confidence level of a network's individual predictions and its overall accuracy (Dawid, 1982; Degroot & Fienberg, 1983); for poorly calibrated networks, an incorrect prediction might have high confidence. *We argue that conventional pseudo-labeling based methods achieve poor results because poor network calibration produces incorrectly pseudo-labeled samples, leading to noisy training and poor generalization.* To remedy this, we empirically study the relationship between output prediction uncertainty and calibration. We find that selecting predictions with low uncertainty greatly reduces the effect of poor calibration, improving generalization.

Motivated by this, we propose an uncertainty-aware pseudo-label selection (UPS) framework that leverages the prediction uncertainty to guide the pseudo-label selection procedure. We believe pseudo-labeling has been impactful due to its simplicity, generality, and ease of implementation; to this end, our proposed framework attempts to maintain these benefits, while addressing the issue of calibration to drastically improve PL performance. UPS does not require modality-specific augmentations and can leverage most uncertainty estimation methods in its selection process. Furthermore, the proposed framework allows for the creation of negative pseudo-labels (i.e. labels which specify the absence of specific classes). If a network predicts the absence of a class with high confidence and high certainty, then a negative label can be assigned to that sample. This generalization is beneficial for both single-label and multi-label learning. In the single-label case, networks can use these labels for negative learning (Kim et al., 2019)[1]; in the multi-label case, class presence is independent so both positive and negative labels are necessary for training.

Our key contributions include the following: (1) We introduce UPS, a novel uncertainty-aware pseudo-label selection framework which greatly reduces the effect of poor network calibration on the pseudo-labeling process, (2) While prior SSL methods focus on single-label classification, we generalize pseudo-labeling to create negative labels, allowing for negative learning and multi-label classification, and (3) Our comprehensive experimentation shows that the proposed method achieves strong performance on commonly used benchmark datasets CIFAR-10 and CIFAR-100. In addition, we highlight our method's flexibility by outperforming previous state-of-the-art approaches on the video dataset, UCF-101, and the multi-label Pascal VOC dataset.

## 2 RELATED WORKS

Semi-supervised learning is a heavily studied problem. In this work, we mostly focus on pseudo-labeling and consistency regularization based approaches as currently, these are the dominant approaches for SSL. Following (Berthelot et al., 2019), we refer to the other SSL approaches for interested readers which includes: "transductive" models (Gammerman et al., 1998; Joachims, 1999; 2003), graph-based methods (Zhu et al., 2003; Bengio et al., 2006; Liu et al., 2019), generative modeling (Belkin & Niyogi, 2002; Lasserre et al., 2006; Kingma et al., 2014; Pu et al., 2016). Furthermore, several recent self-supervised approaches (Grill et al., 2020; Chen et al., 2020b; Caron et al., 2020), have shown strong performance when applied to the SSL task. For a general overview of SSL, we point to (Chapelle et al., 2010; Zhu, 2005).

**Pseudo-labeling** The goal of pseudo-labeling (Lee, 2013; Shi et al., 2018) and self-training (Yarowsky, 1995; McClosky et al., 2006) is to generate pseudo-labels for unlabeled samples with a model trained on labeled data. In (Lee, 2013), pseudo-labels are created from the predictions of a trained neural network. Pseudo-labels can also be assigned to unlabeled samples based on neighborhood graphs (Iscen et al., 2019). Shi et al. (2018) extend the idea of pseudo-labeling by incorporating confidence scores for unlabeled samples based on the density of a local neighborhood. Inspired by noise correction work (Yi & Wu, 2019), Wang & Wu (2020) attempt to update the pseudo-labels through an optimization framework. Recently, (Xie et al., 2019) show self-training can be used to improve the performance of benchmark supervised classification tasks. A concurrent

---

[1]The motivations for using negative learning (NL) in this work differs greatly from Kim et al. (2019). In this work, NL is used to incorporate more unlabeled samples into training and to generalize pseudo-labeling to the multi-label classification setting, whereas Kim et al. (2019) use negative learning primarily to obtain good network initializations to learn with noisy labels. Further discussion about NL can be found in Appendix K.

work (Haase-Schutz et al., 2020) partitions an unlabeled dataset and trains re-initialized networks on each partition. They use previously trained networks to filter the labels used for training newer networks. However, most of their experiments involve learning from noisy data. Although previous pseudo-labeling based SSL approaches are general and domain-agnostic, they tend to under-perform due to the generation of noisy pseudo-labels; our approach greatly reduces noise by minimizing the effect of poor network calibration, allowing for competitive state-of-the-art results.

**Consistency Regularization** The main objective of consistency regularization methods is to obtain perturbation/augmentation invariant output distribution. In (Sajjadi et al., 2016) random max-pooling, dropout, and random data augmentation are used as input perturbations. In (Miyato et al., 2018) perturbations are applied to the input that changes the output predictions maximally. Temporal ensembling (Laine & Aila, 2017) forces the output class distribution for a sample to be consistent over multiple epochs. Tarvainen & Valpola (2017) reformulate temporal ensembling as a teacher-student problem. Recently, the Mixup (Zhang et al., 2018) augmentation, has been used for consistency regularization in (Verma et al., 2019). Several SSL works combine ideas from both consistency regularization and pseudo-labeling (Berthelot et al., 2019; 2020; Zhou et al., 2020). In (Berthelot et al., 2019), pseudo-labels are generated by averaging different predictions of augmented versions of the same sample and the Mixup augmentation is used to train with these pseudo-labels. The authors in (Berthelot et al., 2020) extend this idea by dividing the set of augmentations into strong and weak augmentations. Also, (Zhou et al., 2020) incorporate a time-consistency metric to effectively select time-consistent samples for consistency regularization. The success of recent consistency regularization methods can be attributed to domain-specific augmentations; our approach does not inherently rely on these augmentations, which allows for application to various modalities. Also, our pseudo-labeling method is orthogonal to consistency regularization techniques; therefore, these existing techniques can be applied alongside UPS to further improve network performance.

**Uncertainty and Calibration** Estimating network prediction uncertainty has been a deeply studied topic (Graves, 2011; Blundell et al., 2015; Louizos & Welling, 2016; Lakshminarayanan et al., 2017; Malinin & Gales, 2018; Maddox et al., 2019; Welling & Teh, 2011). In the SSL domain, (Yu et al., 2019; Xia et al., 2020) use uncertainty to improve consistency regularization learning for the segmentation of medical images. A concurrent work (Mukherjee & Awadallah, 2020), selects pseudo-labels predicted by a pretrained language model using uncertainty for a downstream SSL task. One difference between our works is the selection of hard samples. Whereas Mukherjee et al. select a certain amount of hard samples (i.e. those which are not confident or certain) and learn from these using positive learning, we decide to use negative learning on these samples which reduces the amount of noise seen by the network. Zheng & Yang (2020) show strong performance on the domain adaptive semantic segmentation task by leveraging uncertainty. However, to the best of our knowledge, uncertainty has not been used to reduce the effect of poor network calibration in the pseudo-labeling process. In this work, instead of improving the calibration of the network (Guo et al., 2017; Xing et al., 2020), we present a general framework which can leverage most uncertainty estimation methods to select a better calibrated subset of pseudo-labels.

## 3 PROPOSED METHOD

### 3.1 PSEUDO-LABELING FOR SEMI-SUPERVISED LEARNING

**Notation** Let $D_L = \left\{ \left( x^{(i)}, \boldsymbol{y}^{(i)} \right) \right\}_{i=1}^{N_L}$ be a labeled dataset with $N_L$ samples, where $x^{(i)}$ is the input and $\boldsymbol{y}^{(i)} = \left[ y_1^{(i)}, ..., y_C^{(i)} \right] \subseteq \{0, 1\}^C$ is the corresponding label with $C$ class categories (note that multiple elements in $\boldsymbol{y}^{(i)}$ can be non-zero in multi-label datasets). For a sample $i$, $y_c^{(i)} = 1$ denotes that class $c$ is present in the corresponding input and $y_c^{(i)} = 0$ represent the class's absence. Let $D_U = \{x^{(i)}\}_{i=1}^{N_U}$ be an unlabeled dataset with $N_U$ samples, which does not contain labels corresponding to its input samples. For the unlabeled samples, pseudo-labels $\tilde{\boldsymbol{y}}^{(i)}$ are generated. Pseudo-labeling based SSL approaches involve learning a parameterized model $f_\theta$ on the dataset $\tilde{D} = \left\{ \left( x^{(i)}, \tilde{\boldsymbol{y}}^{(i)} \right) \right\}_{i=1}^{N_L + N_U}$, with $\tilde{\boldsymbol{y}}^{(i)} = \boldsymbol{y}^{(i)}$ for the $N_L$ labeled samples.

**Generalizing Pseudo-label Generation**  There are several approaches to create the pseudo-labels $\tilde{\boldsymbol{y}}^{(i)}$, which have been described in Section 2. We adopt the approach where hard pseudo-labels are obtained directly from network predictions. Let $\boldsymbol{p}^{(i)}$ be the probability outputs of a trained network on the sample $x^{(i)}$, such that $p_c^{(i)}$ represents the probability of class $c$ being present in the sample. Using these output probabilities, the pseudo-label can be generated for $x^{(i)}$ as:

$$\tilde{y}_c^{(i)} = \mathbb{1}\left[p_c^{(i)} \geq \gamma\right], \tag{1}$$

where $\gamma \in (0, 1)$ is a threshold used to produce hard labels. Note that conventional single-label pseudo-labeling can be derived from equation 1 when $\gamma = \max_c p_c^{(i)}$. For the multi-label case, $\gamma = 0.5$ would lead to binary pseudo-labels, in which multiple classes can be present in one sample.

## 3.2 Pseudo-label Selection

Although pseudo-labeling is versatile and modality-agnostic, it achieves relatively poor performance when compared to recent SSL methods. This is due to the large number of incorrectly pseudo-labeled samples used during training. Therefore, we aim at reducing the noise present in training to improve the overall performance. This can be accomplished by intelligently selecting a subset of pseudo-labels which are less noisy; since networks output confidence probabilities for class presence (or class absence), we select those pseudo-labels corresponding with the high-confidence predictions.

Let $\boldsymbol{g}^{(i)} = \left[g_1^{(i)}, ..., g_C^{(i)}\right] \subseteq \{0, 1\}^C$ be a binary vector representing the selected pseudo-labels in sample $i$, where $g_c^{(i)} = 1$ when $\tilde{y}_c^{(i)}$ is selected and $g_c^{(i)} = 0$ when $\tilde{y}_c^{(i)}$ is not selected. This vector is obtained as follows:

$$g_c^{(i)} = \mathbb{1}\left[p_c^{(i)} \geq \tau_p\right] + \mathbb{1}\left[p_c^{(i)} \leq \tau_n\right], \tag{2}$$

where $\tau_p$ and $\tau_n$ are the confidence thresholds for positive and negative labels (here, $\tau_p \geq \tau_n$). If the probability score is sufficiently high ($p_c^{(i)} \geq \tau_p$) then the positive label is selected; conversely, a network is sufficiently confident of a class's absence ($p_c^{(i)} \leq \tau_n$), in which case the negative label is selected.

The parameterized model $f_\theta$ is trained on the selected subset of pseudo-labels. For single-label classification, cross-entropy loss is calculated on samples with selected positive pseudo-labels. If no positive label is selected, then negative learning is performed, using negative cross-entropy loss:

$$L_{\text{NCE}}\left(\tilde{\boldsymbol{y}}^{(i)}, \hat{\boldsymbol{y}}^{(i)}, \boldsymbol{g}^{(i)}\right) = -\frac{1}{s^{(i)}} \sum_{c=1}^{C} g_c^{(i)} \left(1 - \tilde{y}_c^{(i)}\right) \log \left(1 - \hat{y}_c^{(i)}\right), \tag{3}$$

where $s^{(i)} = \sum_c g_c^{(i)}$ is the number of selected pseudo-labels for sample $i$. Here, $\hat{\boldsymbol{y}}^{(i)} = f_\theta\left(x^{(i)}\right)$ is the probability output for the model $f_\theta$. For multi-label classification, a modified binary cross-entropy loss is utilized:

$$L_{\text{BCE}}\left(\tilde{\boldsymbol{y}}^{(i)}, \hat{\boldsymbol{y}}^{(i)}, \boldsymbol{g}^{(i)}\right) = -\frac{1}{s^{(i)}} \sum_{c=1}^{C} g_c^{(i)} \left[\tilde{y}_c^{(i)} \log \left(\hat{y}_c^{(i)}\right) + \left(1 - \tilde{y}_c^{(i)}\right) \log \left(1 - \hat{y}_c^{(i)}\right)\right]. \tag{4}$$

In both cases, the selection of high confidence pseudo-labels removes noise during training, allowing for improved performance when compared to traditional pseudo-labeling.

## 3.3 Uncertainty-Aware Pseudo-label Selection

Although confidence-based selection reduces pseudo-label error rates, the poor calibration of neural networks renders this solution insufficient - in poorly calibrated networks, incorrect predictions can have high confidence scores. Since calibration can be interpreted as a notion of a network's overall prediction uncertainty (Lakshminarayanan et al., 2017), the question then arises: *Is there a relationship between calibration and individual prediction uncertainties?* To answer this question, we empirically analyze the relationship between the Expected Calibration Error (ECE) score[2] (Guo

---

[2] An in-depth description of the ECE score is included in section G of the Appendix.

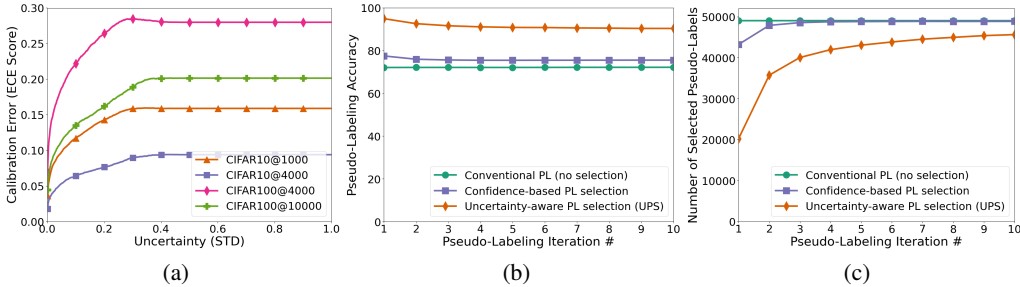

Figure 1: (a) The relationship between prediction uncertainty and expected calibration error (ECE). In all datasets, as the uncertainty of the selected pseudo-labels decreases, the ECE of that selected subset decreases. (b) Comparison of pseudo-label selection accuracy between conventional pseudo-labeling (PL), confidence-based selection (Confidence PL), and UPS. (c) Comparison of the number of selected pseudo-labels between conventional pseudo-labeling (PL), confidence-based selection (Confidence PL), and UPS. Although UPS initially selects a smaller set of pseudo-labels, by the final pseudo-labeling iterations it incorporates the majority of pseudo-labels in training, while maintaining a higher pseudo-labeling accuracy (as seen in (b)). Figures (b) and (c) are generated from the CIFAR-10 dataset with 1000 labels.

et al., 2017) and output prediction uncertainties. Figure 1a illustrates the direct relationship between the ECE score and output prediction uncertainty; when pseudo-labels with more certain predictions are selected, the calibration error is greatly reduced for that subset of pseudo-labels. Therefore, for that subset of labels, a high-confidence prediction is more likely to lead to a correct pseudo-label.

From this observation, we conclude that prediction uncertainties can be leveraged to negate the effects of poor calibration. Thus, we propose an uncertainty-aware pseudo-label selection process: by utilizing both the confidence and uncertainty of a network prediction, a more accurate subset of pseudo-labels are used in training. Equation 2 now becomes,

$$g_c^{(i)} = \mathbb{1}\left[u\left(p_c^{(i)}\right) \leq \kappa_p\right] \mathbb{1}\left[p_c^{(i)} \geq \tau_p\right] + \mathbb{1}\left[u\left(p_c^{(i)}\right) \leq \kappa_n\right] \mathbb{1}\left[p_c^{(i)} \leq \tau_n\right], \qquad (5)$$

where $u(p)$ is the uncertainty of a prediction $p$, and $\kappa_p$ and $\kappa_n$ are the uncertainty thresholds. This additional term, involving $u(p)$, ensures the network prediction is sufficiently certain to be selected. Figure 1b shows that this uncertainty-aware selection process greatly increases pseudo-label accuracy when compared to both traditional pseudo-labeling and confidence-based selection.

### 3.4 LEARNING WITH UPS

First, a neural network $f_{\theta,0}$ is trained on the small labeled set, $D_L$. Once trained, the network generates predictions for all unlabeled data in $D_U$; pseudo-labels are created from these predictions following equation 1. A subset of the pseudo-labels is selected using UPS (equation 5). Next, another network $f_{\theta,1}$, is trained using the selected pseudo-labels as well as the labeled set. This process is continued iteratively until convergence is observed in terms of the number of selected pseudo-labels. Figure 1c illustrates that most pseudo-labels are selected by the end of the training. To limit error propagation in our iterative training and pseudo-labeling process, we generate new labels for all unlabeled samples and reinitialize the neural network after each pseudo-labeling step. The complete training procedure is described in Algorithm 1 in Section B of the Appendix.

## 4 EXPERIMENTAL EVALUATION

**Datasets** To show the versatility of UPS we conduct experiments on four diverse datasets: CIFAR-10, CIFAR-100 (Krizhevsky et al., 2009), Pascal VOC2007 (Everingham et al.), and UCF-101 (Soomro et al., 2012). CIFAR-10 and CIFAR-100 are standard benchmark datasets, with 10 and 100 class categories respectively; both contain $60,000$, $32 \times 32$ images, split into $50,000$ training images and $10,000$ test images. Pascal VOC2007 is a multi-label dataset with $5,011$ training images and $4,952$ test images. It consists of 20 classes and each sample contains between 1 and 6 class categories. We also evaluate our method on the video dataset UCF-101, which contains 101 action classes. We use the standard train/test split with $9,537$ videos for training and $3,783$ videos for testing.

Table 1: Error rate (%) on the CIFAR-10 and CIFAR-100 test set. Methods with † are pseudo-labeling based, whereas others are consistency regularization methods.

| Method | CIFAR-10 | | CIFAR-100 | |
| --- | --- | --- | --- | --- |
| | 1000 labels | 4000 labels | 4000 labels | 10000 labels |
| DeepLP[†] | $22.02 \pm 0.88$ | $12.69 \pm 0.29$ | $46.20 \pm 0.76$ | $38.43 \pm 1.88$ |
| TSSDL[†] | $21.13 \pm 1.17$ | $10.90 \pm 0.23$ | - | - |
| MT | $19.04 \pm 0.51$ | $11.41 \pm 0.25$ | $45.36 \pm 0.49$ | $36.08 \pm 0.51$ |
| MT + DeepLP | $16.93 \pm 0.70$ | $10.61 \pm 0.28$ | $43.73 \pm 0.20$ | $35.92 \pm 0.47$ |
| ICT | $15.48 \pm 0.78$ | $7.29 \pm 0.02$ | - | - |
| DualStudent | $14.17 \pm 0.38$ | $8.89 \pm 0.09$ | - | $32.77 \pm 0.24$ |
| R2-D2 | - | - | - | $32.87 \pm 0.51$ |
| MixMatch | - | 6.84 | - | - |
| UPS[†] | $\mathbf{8.18 \pm 0.15}$ | $\mathbf{6.39 \pm 0.02}$ | $\mathbf{40.77 \pm 0.10}$ | $\mathbf{32.00 \pm 0.49}$ |

## 4.1 IMPLEMENTATION DETAILS

For CIFAR-10 and CIFAR-100 experiments, we use the CNN-13 architecture that is commonly used to benchmark SSL methods (Oliver et al., 2018a; Luo et al., 2018). For the Pascal VOC2007 experiments, we use ResNet-50 (He et al., 2016) with an input resolution of $224 \times 224$. Finally, for UCF-101, we follow the experimental setup of (Jing et al., 2020) by using 3D ResNet-18 (Hara et al., 2018) with a resolution of $112 \times 112$ and 16 frames. For all experiments, we set a dropout rate of 0.3. We use SGD optimizer with an initial learning rate of 0.03 and cosine annealing (Loshchilov & Hutter, 2017) for learning rate decay. We set the confidence thresholds $\tau_p = 0.7$ and $\tau_n = 0.05$ for all experiments, except on the Pascal VOC dataset, where $\tau_p = 0.5$ as it is a multi-label dataset and strict confidence threshold significantly reduces the number of positive pseudo-labels for difficult classes. Furthermore, for the uncertainty thresholds we use $\kappa_p = 0.05$ and $\kappa_n = 0.005$ for all experiments. The threshold, $\gamma$, used to generate the pseudo-labels is set to $\gamma = \max_c p_c^{(i)}$ in single-label experiments and $\gamma = 0.5$ in multi-label experiments.

Our framework can utilize most uncertainty estimation method for selecting pseudo-labels (see section 5 for experiments with different uncertainty estimation methods). Unless otherwise stated, we use MC-Dropout (Gal & Ghahramani, 2016) to obtain an uncertainty measure by calculating the standard deviation of 10 stochastic forward passes. To further reduce the calibration error and to make the negative pseudo-label selection more robust, we perform temperature scaling to soften the output predictions - we set $T = 2$ (Guo et al., 2017). Moreover, networks trained on datasets with few classes tend to be biased towards easy classes in the initial pseudo-labeling iterations. This leads to the selection of more pseudo-labels for these classes, causing a large class imbalance. To address this issue, we balance the number of selected pseudo-labels for all classes; this constraint is removed after 10 pseudo-labeling iterations in CIFAR-10 and after 1 pseudo-labeling iteration in Pascal VOC. The effect of this balancing is presented in section F of the Appendix.

## 4.2 RESULTS

**CIFAR-10 and CIFAR-100** We conduct experiments on CIFAR-10 for two different labeled set sizes (1000 and 4000 labels), as well as on CIFAR-100 with labeled set sizes of 4000 and 10000 labels. For a fair comparison, we compare against methods which report results using the CNN-13 architecture: DeepLP (Iscen et al., 2019), TSSDL (Shi et al., 2018), MT (Tarvainen & Valpola, 2017), MT + DeepLP, ICT (Verma et al., 2019), DualStudent (Ke et al., 2019), and MixMatch (Berthelot et al., 2019). The results reported in Table 1, are the mean and standard deviation from experiments across three different random splits. We achieve comparable results to the state-of-the-art holistic method MixMatch (Berthelot et al., 2019) for CIFAR-10 with 4000 labels, with a $0.45\%$ improvement. *Our CIFAR-10 experiment with 1000 labeled samples outperforms previous methods which use the CNN-13 architecture*. Also, we *outperform* previous methods in the CIFAR-100 experiments.

Table 2: Error rate (%) on CIFAR-10 with different backbones Wide ResNet-28-2 (WRN) and Shake-Shake (S-S).

| Method | Backbone | Labels | |
| --- | --- | --- | --- |
| | | 1000 | 4000 |
| MixMatch | WRN | 7.75 | 6.24 |
| MixMatch | S-S | - | 4.95 |
| ReMixMatch | WRN | 5.73 | 5.14 |
| TC-SSL | WRN | 6.15 | 5.07 |
| R2-D2 | S-S | - | 5.72 |
| UPS | WRN | 7.95 | 6.42 |
| UPS | S-S | - | 4.86 |

Table 3: Accuracy (%) on the UCF-101 test set. Methods with * use scores reported in (Jing et al., 2020).

| Method | 20% labeled | 50% labeled |
|---|---|---|
| Supervised | 33.5 | 45.6 |
| MT* | 36.3 | 45.8 |
| PL* | 37.0 | 47.5 |
| S4L* | 37.7 | 47.9 |
| UPS | **39.4** | **50.2** |

Table 4: mAP scores on the Pascal VOC2007 test set.

| Method | 10% labeled | 20% labeled |
|---|---|---|
| Supervised | $18.36 \pm 0.65$ | $28.84 \pm 1.68$ |
| PL | $27.44 \pm 0.55$ | $34.84 \pm 1.88$ |
| MixMatch | $29.57 \pm 0.78$ | $37.02 \pm 0.97$ |
| MT | $32.55 \pm 1.48$ | $39.62 \pm 1.66$ |
| UPS | $\mathbf{34.22 \pm 0.79}$ | $\mathbf{40.34 \pm 0.08}$ |

We present additional results on CIFAR-10 with better backbone networks (Wide ResNet 28-2 (Zagoruyko & Komodakis, 2016) and Shake-Shake (Gastaldi, 2017)) in Table 2, and compare with methods: MixMatch, ReMixMatch (Berthelot et al., 2020), TC-SSL (Zhou et al., 2020), R2-D2 (Wang & Wu, 2020). We find that UPS is not backbone dependent, and achieves further performance improvements when a stronger backbone is used.

**UCF-101** For our UCF-101 experiments, we evaluate our method on using 20% and 50% of the training data as the labeled set. A comparison of our method and several SSL baselines is reported in Table 3. The results reported for the baselines PL (Lee, 2013), MT (Tarvainen & Valpola, 2017), and S4L (Zhai et al., 2019) are obtained from (Jing et al., 2020), as it uses the same network architecture and a similar training strategy. We do not compare directly with (Jing et al., 2020), since they utilize a pretrained 2D appearance classifier which makes it an unfair comparison. *Even though none of the reported methods, including UPS, are developed specifically for the video domain, UPS outperforms all SSL methods.* Interestingly, both pseudo-labeling (PL) and UPS achieve strong results, when compared to the consistency-regularization based method, MT (Tarvainen & Valpola, 2017) .

**Pascal VOC2007** We conduct two experiments with $10\%$ (500 samples) and $20\%$ (1000 samples) of the train-val split as the labeled set. Since *there is no prior work on multi-label semi-supervised classification*, we re-implement three methods: Pseudo-labeling (PL) (Lee, 2013), MeanTeacher (MT) (Tarvainen & Valpola, 2017), and MixMatch (Berthelot et al., 2019). For a fair comparison, we use the same network architecture and similar training strategy for all baselines. Table 4 shows UPS outperforming all methods with $1.67\%$ and $0.72\%$ improvements when using $10\%$ and $20\%$ of the labeled data, respectively. One reason why UPS and MT performs strongly in multi-label classification that neither approach has a single-label assumption; meanwhile, recent SSL methods like MixMatch and ReMixMatch are designed for single-label classification (e.g. temperature sharpening assumes a softmax probability distribution), which make them difficult to apply to multi-label datasets[3].

## 4.3 ABLATIONS

We present an ablation study to measure the contribution of the method's different components. We run experiments on CIFAR-10 with 1000 and 4000 labeled samples. Table 5 displays the results from our ablation study. Standard pseudo-labeling (UPS, no selection) slightly improves upon the supervised baseline; this is caused by the large number of incorrect pseudo-labels present in training. Through confidence-based selection (UPS, no uncertainty-aware (UA) selection), many incorrect pseudo-label are ignored, leading to $6.1\%$ and $3.71\%$ error rate reductions for 1000 and 4000

Table 5: Ablation Study on CIFAR-10 dataset (Error Rate (%)). UPS with no uncertainty-aware (UA) selection, selects using only confidence-based criteria.

| Method | 1000 labels | 4000 labels |
|---|---|---|
| Supervised | 27.66 | 16.65 |
| UPS, no selection | 22.60 | 12.94 |
| UPS, no UA | 16.50 | 10.02 |
| UPS, no UA (Cal.) | 13.68 | 8.09 |
| UPS, no NL | 9.46 | 6.64 |
| UPS, full method | 8.14 | 6.36 |

labels respectively. When MC-Dropout is used to improve network calibration (UPS, no UA (Cal.)), there is a slight improvement; however, this calibration is not adequate to produce sufficiently accurate pseudo-labels. Hence, by including uncertainty into the selection process, a further improvement of $5.54\%$ is observed for 1000 samples and $1.73\%$ for 4000 samples. We also find that negative learning (NL) is beneficial in both experimental setups. By incorporating more unlabeled samples in training (i.e. samples which do not have a confident and certain positive label), NL leads to $1.32\%$ and $0.32\%$ error rate reductions for 1000 and 4000 samples respectively.

---

[3]Details on how MixMatch was adapted for this experiment can be found in section H of the Appendix.

## 4.4 ANALYSIS

**Robustness to Hyperparameters** Our framework introduces new threshold hyperparameters $\tau$ and $\kappa$. Following (Oliver et al., 2018b) we do not "over-tweak" the hyperparameters - we select thresholds based on a CIFAR-10 validation set of 1000 samples[4]. Although our experiments set $\kappa_p = 0.05$, we find that UPS is relatively robust to this hyperparameter. Figure 2 shows the test error produced when using various uncertainty thresholds. We find that using $\kappa_p < 0.1$ leads to comparable performance, and further increases of the threshold lead to predictable performance drops (as the threshold increases, more noisy pseudo-labels are selected leading to higher test error).

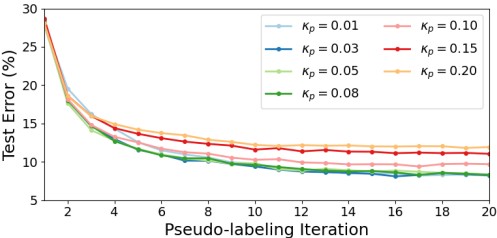

Figure 2: Robustness to uncertainty threshold. Thresholds below 0.1 lead to similar test error on CIFAR-10 (1000 labels), showing that UPS is not reliant on a single threshold.

Once the uncertainty threshold is selected, confidence thresholds $\tau_p > 0.5$ also lead to similar performance. UPS requires little hyperparameter tuning: although our thresholds were selected using the CIFAR-10 validation set, these same thresholds were used successfully on the other datasets (CIFAR-100, UCF-101, and Pascal VOC) and splits.

**UPS vs Confidence-based PL** To investigate the performance difference between our approach and conventional pseudo-labeling, we analyse the subset of labels which UPS selects. Our analysis is performed on CIFAR-10 with 1000 labeled samples. Both confidence based selection and UPS improve pseudo-labeling accuracy, while still utilizing the majority of samples by the end of the training, as seen in Figures 1b and 1c. Specifically, after 10 pseudo-labeling iterations, UPS selects about 93.12% of the positive pseudo-labels with an accuracy of 90.29%. Of the remaining 3371 samples which do not have a selected positive pseudo-label, over 88% of them are still used in training through negative learning: negative pseudo-labels are selected for 2988 samples, with an average of 6.57 negative pseudo-labels per sample. Even though confidence-based selection improves upon conventional PL in terms of label accuracy (77.44% vs. 72.03% in the initial pseudo-labeling step), it is insufficient to achieve strong overall performance. UPS overcomes this problem by initially selecting a smaller, more accurate subset (20217 positive labels with an accuracy of 94.87%), and gradually increasing the number of selected pseudo-labels while maintaining high accuracy.

## 5 DISCUSSION

**Uncertainty Estimation** UPS is a general framework, it does not depend on a particular uncertainty measure. In our experiments, we use MC-Dropout (Gal & Ghahramani, 2016) to obtain the uncertainty measure. Ideally, approximate Bayesian inference methods (Graves, 2011; Blundell et al., 2015; Louizos & Welling, 2016) can be used to obtain prediction uncertainties; however, Bayesian NNs are computationally costly

Table 6: Comparison of methods for uncertainty estimation on CIFAR-10 (1000 labels) (Error Rate (%))

| Method | 1000 labels | 4000 labels |
|---|---|---|
| MC-Dropout | 8.14 | 6.36 |
| MC-SpatialDropout | 8.28 | 6.60 |
| MC-DropBlock | 9.76 | 7.50 |
| DataAug | 8.28 | 6.72 |

and more difficult to implement than non-Bayesian NNs. Instead, methods like (Wan et al., 2013; Lakshminarayanan et al., 2017; Tompson et al., 2015; Ghiasi et al., 2018) can be used without extensive network modification to obtain an uncertainty measure directly (or through Monte Carlo sampling) that can easily be incorporated into UPS. To this end, we evaluate UPS using three other uncertainty estimation methods using MC sampling with SpatialDropout (Tompson et al., 2015) and DropBlock (Ghiasi et al., 2018), as well as random data augmentation (DataAug). The experimental settings are described in Section D of the Appendix. Without using uncertainty estimation method specific hyperparameters, we find that UPS achieves comparable results when using any of these methods, as seen in Table 6.

---

[4]Additional information on hyperparameter selection can be found in section I of the Appendix.

**Data Augmentation in SSL**    One major advantage of UPS over recent state-of-the-art consistency regularization based SSL methods is that it does not inherently rely on domain-specific data augmentations. For different data modalities like video, text, and speech it is not always possible to obtain a rich set of augmentations. This is evident in Table 3, where both standard pseudo-labeling approach (Lee, 2013) and UPS outperform MT (Tarvainen & Valpola, 2017) on the video dataset, UCF-101. Recent state-of-the-art SSL methods, like (Berthelot et al., 2020; Sohn et al., 2020), divide the augmentation space into the sets of strong and weak augmentations, which is possible for the image domain as it has many diverse augmentations. However, it is not straightforward to extend these methods to other data modalities; for instance, in the video domain, the two dominant augmentations are spatial crop and temporal jittering, which are difficult to divide into strong and weak subcategories. Moreover, the Mixup (Zhang et al., 2018) data augmentation is used in several recent SSL methods (Verma et al., 2019; Berthelot et al., 2019; 2020); it achieves strong results on single-label data, but extensions to multi-label classification have been ineffective (Wang et al., 2019). Although using augmentations during training improves network performance (see section E in the Appendix), the existence of domain-specific augmentations is not a prerequisite for UPS.

## 6  CONCLUSION

In this work, we propose UPS, an uncertainty-aware pseudo-label selection framework that maintains the simplicity, generality, and ease of implementation of pseudo-labeling, while performing on par with consistency regularization based SSL methods. Due to poor neural network calibration, conventional pseudo-labeling methods trained on a large number of incorrect pseudo-labels result in noisy training; our pseudo-label selection process utilizes prediction uncertainty to reduce this noise. This results in strong performance on multiple benchmark datasets. The unique properties of UPS are that it can be applied to multiple data modalities and to both single-label and multi-label classification. We hope that in the future, the machine learning community will focus on developing general SSL algorithms which do not have inherent limitations like domain-specific augmentations and single-label assumptions.

**ACKNOWLEDGEMENTS**    This research is based upon work supported by the Office of the Director of National Intelligence (ODNI), Intelligence Advanced Research Projects Activity (IARPA), via IARPA RD Contract No. D17PC00345. The views and conclusions contained herein are those of the authors and should not be interpreted as necessarily representing the official policies or endorsements, either expressed or implied, of the ODNI, IARPA, or the U.S. Government. The U.S. Government is authorized to reproduce and distribute reprints for Governmental purposes notwithstanding any copyright annotation thereon.

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

## A  APPENDIX

In the appendix we include the following: training procedure of UPS (section B), additional experiments on Pascal VOC, UCF-101, and CIFAR-10 (section C), implementation details of the uncertainty estimation (section D), experiments with additional data augmentation (section E), analysis of the effects of class balancing in pseudo-label selection (section F), additional details on ECE (section G), details pertaining to the use of MixMatch in multi-label classification (section H), additional details about hyperparameter selction (section I), and some qualitative results (section L).

## B  UPS TRAINING PROCEDURE

The training procedure for our proposed UPS framework is described in Algorithm 1.

---
**Algorithm 1** The proposed method takes a set of labeled data, $D_L$, and a set of unlabeled data, $D_U$, and returns a trained model, $f_\theta$, using samples from both $D_L$ and $D_U$

---
1: Train a network, $f_{\theta,0}$, using the samples from $D_L$.
2: **for** $i = 1..\text{MaxIterations}$ **do**     ▷ Repeats until convergence
3:     Pseudo-label $D_U$ using $f_{\theta,i-1}$     ▷ Equation 1
4:     $D_{\text{selected}} \leftarrow$ Select pseudo-labels using UPS     ▷ Equation 5
5:     $\tilde{D} \leftarrow D_L \cup D_{\text{selected}}$
6:     Initialize new network $f_{\theta,i}$
7:     Train $f_{\theta,i}$ using the samples from $\tilde{D}$.  ▷ Using cross-entropy loss or losses in Equations 4-5
8:     $f_\theta \leftarrow f_{\theta,i}$
9: **return** $f_\theta$

---

## C  ADDITIONAL RESULTS

### C.1  FULLY SUPERVISED CLASSIFICATION SCORES

For Pascal VOC2007 and UCF-101 datasets the fully supervised classification scores are presented in Table 7; these scores can be viewed as an upper-bound for our method, as our method trains in a supervised manner on the pseudo-labeled samples and on a percentage of the labeled data. These results should not be confused with the "Supervised" baseline in Tables 4 and 3 of our paper, which involve training with *only* the listed percentage of data. With UPS we obtain a mAP of 40.34 and 34.22 when 20% and 10% of the Pascal VOC2007 training-validation data is available respectively. Using all labeled data, a fully supervised network achieves 52.62% mAP with the ResNet-50 network.

For UCF-101, a fully supervised 3D ResNet-18 achieves 50.4% accuracy. Even with only $50\%$ of the labeled data, UPS is able to achieve a similar accuracy (50.2%).

Table 7: Fully supervised classification score on Pascal VOC2007 and UCF-101 dataset. The metrics used for each dataset are mAP and accuracy, respectively.

| Dataset | UPS, 10% data | UPS, 20% data | UPS, 50% data | Supervised, all data |
|---|---|---|---|---|
| Pascal VOC2007 | $34.22 \pm 0.79$ | $40.34 \pm 0.08$ | - | 52.62 |
| UCF-101 | - | 39.4 | 50.2 | 50.4 |

## C.2 CIFAR-10 RESULTS

We conduct experiments on the CIFAR-10 dataset with 250 and 500 labeled examples. The results are presented in table 8. We achieve similar results (within 1.5%) to MixMatch (Berthelot et al., 2019) when using CNN-13 on 250 labels. Most other pseudo-labeling based semi-supervised learning methods do not present results using 250 and 500 labeled samples.

Table 8: Error rates on CIFAR-10 dataset with very few training samples.

| Method | 250 labels | 500 labels |
|---|---|---|
| UPS | $15.90 \pm 0.61$ | $10.64 \pm 0.18$ |

## D UNCERTAINTY ESTIMATION: IMPLEMENTATION DETAILS

The proposed UPS framework can leverage most uncertainty estimation methods to select a better calibrated subset of pseudo-labels. In Table 6 of the main text we show that UPS performs comparably when uncertainty is estimated using (Tompson et al., 2015; Ghiasi et al., 2018) through Monte Carlo sampling during inference time. For MC-SpatialDropout we set the dropout rate to 0.3 and performed 10 stochastic forward passes to obtain an uncertainty measure from the standard deviation of the output probabilities. We follow the same MC sampling strategy with MC-DropBlock. Following (Ghiasi et al., 2018) we set the keep probability to 0.9 for the experiment with MC-DropBlock. For estimating uncertainty with random data augmentation we perform 10 forward passes while performing random crop and random horizontal flip.

## E DATA AUGMENTATION

Since it is common practice to use data augmentation on image datasets, we use RandAugment (Cubuk et al., 2019) for experiments on CIFAR-10, CIFAR-100, and Pascal VOC2007 datasets. For UCF-101 dataset experiments, we use random crop and temporal jittering following (Jing et al., 2020). The Mixup augmentation (Zhang et al., 2018) has become widely used in both supervised and semi-supervised classification. We test how the addition of this powerful augmentation technique could improve UPS on single-label classification. As the extension of Mixup to negative learning is non-trivial, we do not include negative learning in this experiment. For Mixup, we set the hyper-parameter $\alpha$ to 0.50. Since the output prediction with mixup augmentation is better calibrated (Thulasidasan et al., 2019) we use relaxed thresholds for $\tau_p$ (0.50) and $\kappa_p$ (0.10). The results are presented in table 9. As expected the improved augmentation leads to an improvement: it achieves a 2.16% reduction in error when compared to UPS without negative learning on the 1000 label experiment. Notably, UPS+Mixup even outperforms UPS with negative learning.

Since our method is not inherently reliant on specific data augmentations, we run additional experiments on CIFAR-10, with no input augmentations. The results are shown in the table 10. Our method achieves an error rate of 28.14% and 14.98% for 1000 and 4000 labels respectively. This is a respectable score that improves upon other SSL methods $\Pi$ model (Laine & Aila, 2017) and Mean Teacher (Tarvainen & Valpola, 2017) in the same experimental setting (i.e. no data augmentations).

Table 9: The effect of using the Mixup augmentation with UPS on CIFAR-10 dataset with 1000 labels.

| Method | Error Rate (%) |
|---|---|
| UPS, without negative learning | 9.46 |
| UPS, with negative learning | 8.14 |
| UPS+Mixup, without negative learning | 7.30 |

Table 10: Error rates on the CIFAR-10 test set with no input augmentations used during training.

| Method | 1000 labels | 4000 labels |
|---|---|---|
| Π model | 32.18 | 17.08 |
| MT | 30.62 | 17.74 |
| UPS | 28.14 | 14.98 |

## F EFFECT OF CLASS BALANCING

In our experiments, we observe that generating pseudo-labels with only a limited number of labeled samples is difficult especially for the datasets with a small number of classes - CIFAR-10 and Pascal VOC2007. For these datasets, when the number of training samples is limited, the network tends to be biased towards easy classes which leads to class imbalance during the pseudo-label selection; this is specifically true for the initial pseudo-labeling iterations. Table 11 presents the number of selected pseudo-labels for each class of CIFAR-10 during the first pseudo-labeling iteration. For 1000 labeled samples, there is an imbalance between the number of selected pseudo-labels for each class; the cat class has only 1065 pseudo-labels selected whereas the automobile class has 3734 selected pseudo-labels, leading to an imbalance ratio of 3.5. For the CIFAR-10 experiment with 4000 labeled samples the imbalance ratio is not that severe but it's still 2.35. To address this issue we make the pseudo-label selection class balanced for CIFAR-10 for the first 10 pseudo-labeling iterations. Even though Pascal VOC2007 itself is not a class balanced dataset, it has only 20 object classes and training with limited data results in a trained classifier biased towards easy classes. Therefore, for the first iteration of pseudo-label selection, we enforce class balancing for the Pascal VOC2007 dataset, which results in improvement. The results with and without class balancing are presented in table 12. It is evident that class balancing is more impactful when fewer labeled samples are used in training.

Table 11: Number of selected pseudo-labels from each class of CIFAR-10.

| Class ID and Name | 1000 labels | 4000 labels |
|---|---|---|
| 0 (airplane) | 2707 | 3274 |
| 1 (automobile) | 3734 | 3900 |
| 2 (bird) | 1929 | 2377 |
| 3 (cat) | 1065 | 1658 |
| 4 (deer) | 2145 | 2898 |
| 5 (dog) | 1924 | 2273 |
| 6 (frog) | 3224 | 3468 |
| 7 (horse) | 3266 | 3403 |
| 8 (ship) | 3083 | 3538 |
| 9 (truck) | 3214 | 3920 |

## G EXPECTED CALIBRATION ERROR (ECE) COMPUTATION

In our work, we analyse the effect of prediction uncertainty and network calibration. A standard metric for measuring network calibration is Expected Calibration Error (ECE) (Guo et al., 2017; Xing et al., 2020) score,

$$ECE = \sum_{l=1}^{L} \frac{1}{|D|} | \sum_{x^{(i)} \in I_l} \max_{c} \hat{y}_c^{(i)} - \sum_{x^{(i)} \in I_l} \mathbb{1} \left[ \arg\max_{c} \hat{y}_c^{(i)} = \arg\max_{c} \tilde{y}_c^{(i)} \right] |, \qquad (6)$$

Table 12: Performance on the CIFAR-10 and Pascal VOC2007 test sets.

| Method | CIFAR-10 (accuracy) | | Pascal VOC2007 (mAP) | |
|---|---|---|---|---|
| | 1000 labels | 4000 labels | 10% labeled | 20% labeled |
| UPS, with class balance | 91.86 | 93.64 | 34.72 | 40.33 |
| UPS, without class balance | 88.77 | 93.14 | 31.88 | 40.06 |

where the confidence predictions on dataset $D$ are partitioned into $L$ equally-spaced bins. $I_l$ are the samples present in a particular bin $l$. The discrepancy between the average confidence and average accuracy gives the calibration gap of each bin. The average over the calibration gap of all the bins results in ECE score. In our ECE score calculation we have set $L = 15$. Conventionally, ECE is calculated over an entire test set. In our case, we select a subset of unlabeled samples based on their prediction uncertainty and calculate the ECE on this subset. As shown in the main text, we find that as the prediction uncertainty decreases, the ECE score tends to decrease. This implies that neural networks tend to be more calibrated on samples for which it has a lower uncertainty.

## H MixMatch for Multi-Label Classification

MixMatch (Berthelot et al., 2019) is a recent popular SSL method that performs well in the single-label case. We find that this performance does not transfer to the multi-label case. For implementing MixMatch for multi-label classification in the Pascal VOC2007 dataset we use the default parameters mentioned in their paper. We set the value of $\alpha$ to be $0.75$ and we use $K = 2$. Label sharpening for single-label predictions defined below, cannot be applied to multi-label predictions, which assume class independence.

$$\text{Sharpen}(p, T)_i := \frac{p_i^{1/T}}{\sum_{j=1}^{L} p_j^{1/T}}. \tag{7}$$

Sharpening can be performed independently on each output by dividing the logits by a temperature $T$, and applying a sigmoid operation to obtain the probabilities $p$. In our experiments, we found that this did not lead to a significant change in results, so in our main paper the MixMatch experiments on Pascal VOC2007, we report results without label sharpening.

## I Hyperparameter Selection

In this section, we present how hyperparameters are selected in our experiments. As stated in the main text, our hyper-parameters are selected based on a 1000 sample CIFAR-10 validation set. The distribution of pseudo-labels for this validation set can be found in Figures 3a and 3b. The distributions with respect to confidence and uncertainty are skewed toward 1 and 0 respectively. Therefore, selecting confidence and uncertainty thresholds which encompass the majority of pseudo-labels, while maintaining a relatively high accuracy would be sufficient. We find that a similar pattern emerges on the full unlabeled set (Figures 3c and 3d).

On the CIFAR-10 validation set, we select the confidence threshold of $\tau_p = 0.7$ and uncertainty threshold of $\kappa_p = 0.05$ leading to the selection of 531 labels with 92.28% accuracy. We find that once we select this uncertainty threshold, $\kappa_p$, changes in the confidence threshold yields a similar numbers of selected pseudo-labels with similar accuracy. Although, if we select a less strict uncertainty threshold, then changes in the confidence threshold have larger impacts. Since we did not want to over-tweak the confidence threshold from dataset to dataset, we maintained this stricter uncertainty threshold of 0.05 throughout our experiments. However, for any drastically different dataset, we can perform this analysis to obtain a new set of thresholds.

Using a fixed set of hyper-parameters in our experiments demonstrates that UPS can give reasonable performance without dataset specific hyper-parameter tuning; we achieve strong performance on CIFAR-100 and UCF-101 with thresholds obtained from the CIFAR-10 validation set. It is true that a better set of hyper-parameters for a particular dataset can always be found, which is also the case for existing SSL methods (e.g. loss weighting for unlabeled samples and $\alpha$ in the MixUp augmentation

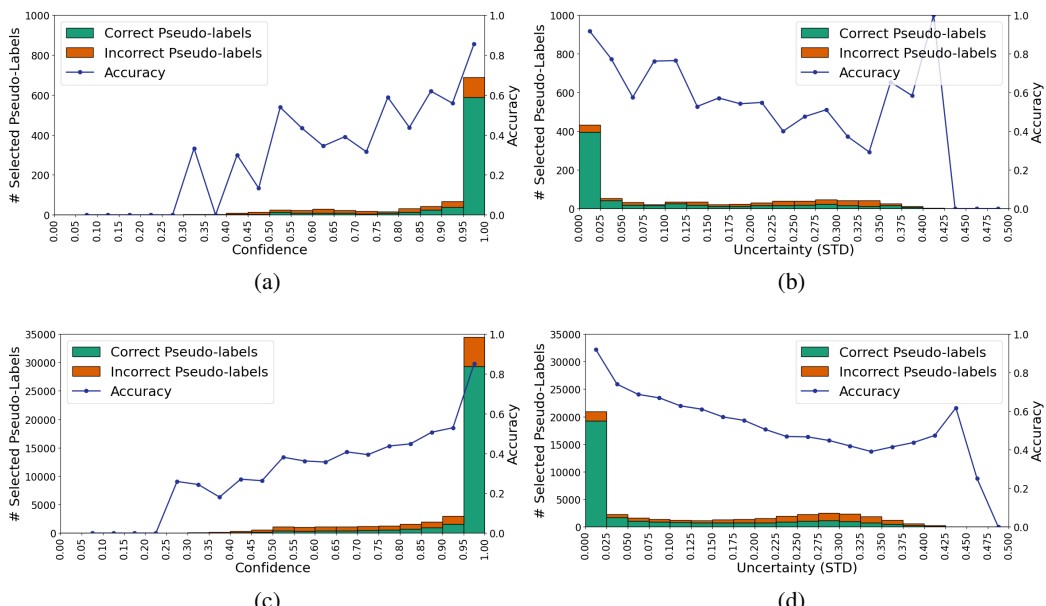

Figure 3: (a) The distribution of correct and incorrect pseudo-labels with respect to the confidence on a 1000 sample CIFAR-10 validation set. (b) The distribution of correct and incorrect pseudo-labels with respect to the uncertainty on a 1000 sample CIFAR-10 validation set. (c) The distribution of correct and incorrect pseudo-labels with respect to the confidence on the set of unlabeled samples. (d) The distribution of correct and incorrect pseudo-labels with respect to the uncertainty on the set of unlabeled samples. Both the full unlabeled set and the small validation have similar skewed confidence and uncertainty distributions.

are both hyper-parameters which can be tuned for improved performance), but the robustness of our method (shown on multiple datasets) and a reasonable hyper-parameter selection strategy enables our method to be applicable to many different datasets.

## J    CALIBRATION AND THRESHOLD SELECTION

We show in our ablations (Table 5) that having a calibrated network with confidence-based thresh-olding achieves better performance than without calibration, when the confidence threshold are the same ($\tau$=0.7). However, we find that even with adjusted thresholds the calibrated network is unable to achieve sufficient pseudo-labeling accuracy to outperform the uncertainty-aware selection. We present the accuracy for the first set of selected pseudo-labels (CIFAR-100, 4000 labels) in Table 13. Increasing the threshold for the calibrated network leads to increased accuracy (which is to be expected), but even with this high threshold of 0.9, it is unable to achieve the 83% selected pseudo-label accuracy of UPS.

Table 13: Pseudo-Labeling accuracy on the CIFAR-100 (4000 labels)

| Method | $\tau_p = 0.7$ | $\tau_p = 0.8$ | $\tau_p = 0.9$ |
|---|---|---|---|
| Conf.-Based Selection | 54.92 | 58.75 | 64.18 |
| Conf.-Based Selection (Cal) | 64.75 | 70.48 | 77.18 |
| UPS | 83.16 | 83.37 | 83.09 |

Based on the trend present in this table, it is feasible that there exists some confidence-based threshold for a network (uncalibrated or calibrated) that could achieve strong pseudo-labeling performance. However, finding such a threshold would be difficult and would not reasonably transfer across datasets. Calibrating the network may make the task simpler, as it moves distribution of confidences, but the use of UPS allows us to find a robust set of thresholds (both confidence and uncertainty) which can be applied across different label splits and datasets.

## K    NEGATIVE LEARNING

There are several distinctions between the use of NL in previous works (Kim et al., 2019) and its use in this work. First, the motivation of using negative labels in this work is to 1) incorporate more unlabeled samples into the training procedure, and 2) to generalize pseudo-labeling for the multi-label classification setting. On the other hand, Kim et al. (2019) use negative learning primarily to obtain good network initializations to learn with noisy labels.

Furthermore, our negative labels are selected in an uncertainty-aware process (equation 5), whereas Kim et al. (2019) initially generates negative labels randomly (NL step) to train a network and then use that network to selectively generate negative labels using confidence scores (SelNL). Their use of selective positive learning (SelPL) also relies on confidence-based positive pseudo-label creation. In our work, we show that relying on confidence-based selection is insufficient, and our proposed uncertainty-aware selection is beneficial for the pseudo-labeling task. In general, our method is not built upon negative learning - we achieve strong performance without NL (see Table 5), but best performance is achieved when the additional negatively pseudo-labeled samples are used during training.

## L    QUALITATIVE RESULTS

In Figure 4 we show some incorrect pseudo-labels obtained from the network trained with 1000 labeled samples from the CIFAR-10 dataset. As the confidence scores for all these images is greater than 0.9, no reasonable confidence based selection criteria would be able to filter out these incorrect predictions. However, UPS easily filters out these incorrect pseudo-labels by leveraging the prediction uncertainties. This demonstrates the adverse effect of poor network calibration in pseudo-labeling, and the benefit of using uncertainty in the selection process.

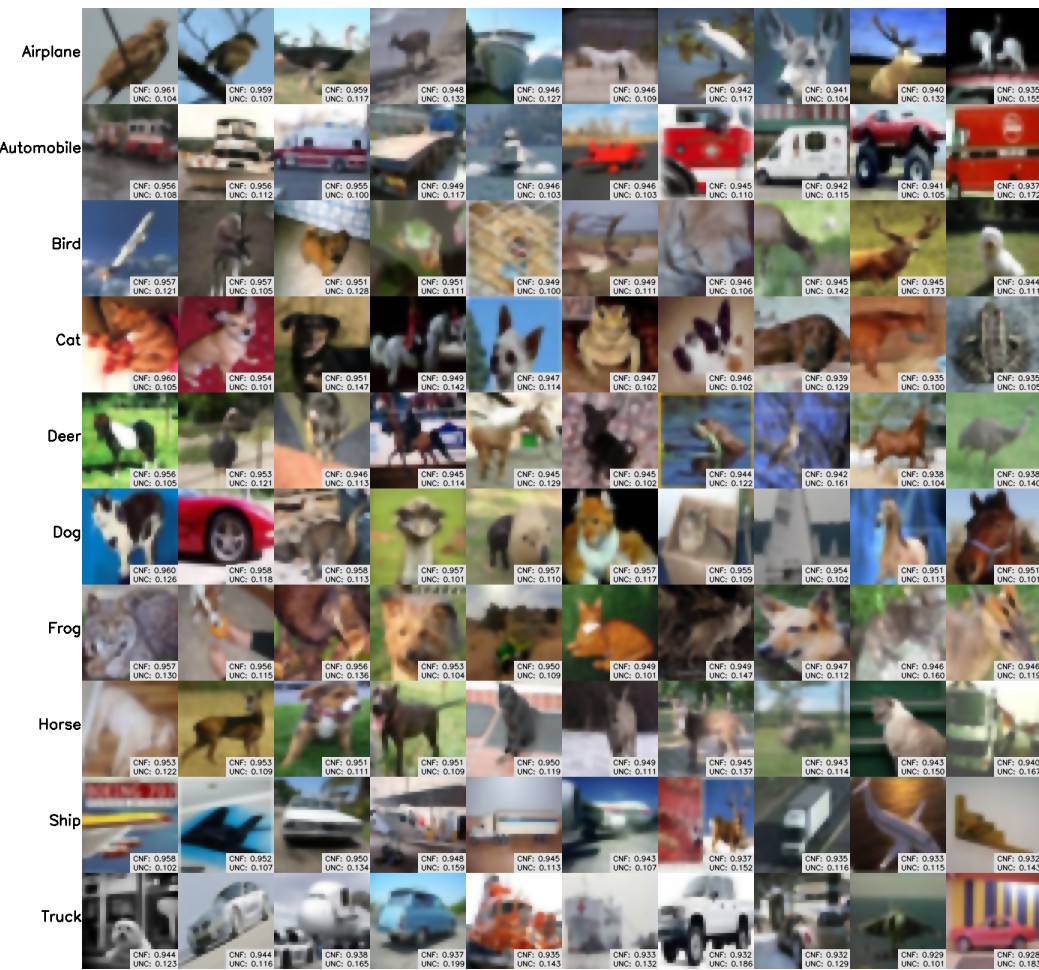

Figure 4: CIFAR-10 samples with incorrect high confidence predictions, that are filtered out by UPS when prediction uncertainty is leveraged.

