# OpenReview forum: "In Defense of Pseudo-Labeling: An Uncertainty-Aware Pseudo-label Selection Framework for Semi-Supervised Learning"
_ICLR.cc/2021/Conference — ICLR 2021 Poster_

### Official Review · AnonReviewer1 · 2020-10-28
**Pseudo-labeling can be on par or better than consistency regularization methods. Method which is **not** based on specific to vision data augmentation.**

**Rating:** 9
**Confidence:** 5

**Review:**

This paper is in defense of simple semi-supervised learning (SSL) with pseudo-labeling (PL): authors demonstrate with experiments on 4 vision datasets (CIFAR-10, CIFAR-100, Pascal VOC and UCF-101) that pseudo-labeling can perform on par with consistency regularization methods. Authors argue that PL doesn't work well because of poor network calibration: because of that the high confident predictions are wrong leading to noisy training and poor generalization. The main contribution of the paper is the usage of prediction uncertainty selection in addition to the confidence-based selection which provides high accuracy of PL used in the further training. Besides this PL is generalized to create negative labels: with this authors perform and show effectiveness of negative learning and multi-label classification. Proposed approach performs in the same ballpark as state-of-the-art methods on CIFAR-10 and CIFAR-100, while it achieves the new state-of-the-art results on video dataset and multi-label task. It is worth to notice that proposed approach is independent from the domain while consistency regularization methods extensively are based on the specific augmentation techniques for the vision/datasets.

Pros:
- Cool idea on predictions uncertainty based selection for pseudo-labeling, no any dependence on domain-specific augmentations for the method
- Analysis of correlation between model calibration and prediction uncertainty
- Analysis of UPS compared to the conventional PL and confidence-based PL
- Ablation study on each component of the method, and dependence on the network architecture
- Well-designed (fair comparison) extensive experiments and comparisons on 4 dataset for multiclass and multi-label classification with better performance than other methods

Cons:
- Absent of large-scale experiments with ImageNet

Comments:
- typo page 3 "would lead to binary psuedo-labels" -> "would lead to binary pseudo-labels"
- "For the multi-label case, $\gamma = 0.5$ would lead to binary pseudo-labels, in which multiple classes can be present in one sample." - this sentence is not clear. If $\gamma = 0.5$ it could be only one or two classes presented in the pseudo-label vector.
- Eq. (2), it is obvious but still please specify that $\tau_p >= \tau_n$
- Figure 1 (b) and (c) - on which data is this analysis done?
- Do authors use the same network $f_{\theta, k}$ on each PL iteration $k$ and just randomly reinitialize it?
- What is $\gamma$ value in experiments?
- For Table 1 would be good to have a clarification on baselines: which are consistency regularization based, which are PL.
- typo in footnote 2 on page 7: add dot at the end of sentence.
- typo page 7 "experimental set-ups." -> "experimental setups.", "labeled samples Both" -> "labeled samples. Both"
- Did authors try experiments on ImageNet too?
- On which data is study in Fig.2 performed?
- typo page 8 "unique in that it can be easily" -> "unique in that: it can be easily" (or any punctuation here)
- For ECE computation is percentile binning used?
- some possible relevant works: https://arxiv.org/pdf/2003.03773.pdf, https://arxiv.org/abs/2006.07733, simCLR v2, https://arxiv.org/abs/2006.09882

It is very well written paper with extensive experiments and ablations (except large-scale experiment), which prove the method efficiency and generalization. Hope, this will push the study of simple SSL approach, pseudo-labeling, with the new competitive results not only in vision but in other domains too.

---

> ### Author Response · Authors · 2020-11-18
> **Response to Reviewer 1**
>
> We would like to thank the reviewer for their feedback and insightful comments. We appreciate the many positive comments relating to our extensive ablations, analysis, and experimentation, as well as the points made about UPS's lack of dependence on domain-specific augmentations. The following are responses to the reviewer's questions and concerns.
>
>
> 1) "For the multi-label case, $\gamma$=0.5 would lead to binary pseudo-labels, in which multiple classes can be present in one sample." - this sentence is not clear. If $\gamma$=0.5 it could be only one or two classes presented in the pseudo-label vector.
>
> We agree that using $\gamma$=0.5 in a single-label dataset would result in only one or two classes present in the pseudo-label vector (since it is applied on the output softmax probability distribution). In the multi-label case, however, the class predictions are assumed to be independent, so a sigmoid activation is used on the logits. This would allow any number of classes to have a predicted probability greater than 0.5, and therefore present in the pseudo-label vector.
>
>
> 2) Figure 1 (b) and (c) - on which data is this analysis done?
>
> This analysis is done on CIFAR-10, 1000 labels.
>
> 3) Do authors use the same network $f_{\theta,1}$ on each PL iteration k and just randomly reinitialize it?
>
> Yes, the same network is used in each iteration. We randomly initialize the network to limit error propagation from previous pseudo-labeling iterations.
>
> 4) What is $\gamma$ value in experiments?
>
> In our single-label experiments, we set gamma to $\gamma = \max_{c} p^{\left(i\right)}_c$ such that the maximal probability prediction is the predicted pseudo-label. For the multi-label experiment on Pascal VOC, we set $\gamma$ to 0.5.
>
> 5) Did authors try experiments on ImageNet too?
>
> In our work, we attempt to evaluate UPS on a variety of different dataset - image (CIFAR-10 and CIFAR-100), multi-label (Pascal VOC), and video (UCF-101) - so we did not include experiments on ImageNet. The study of pseudo-labeling for more large-scale datasets would be an interesting avenue of future work.
>
>
> 6) On which data is study in Fig.2 performed?
>
> This study is done on CIFAR-10, 1000 labels.
>
> 7) For ECE computation is percentile binning used?
>
> When computing ECE, we obtain all prediction confidences and place them in 15 equally spaced bins from 0 to 1 (each with width 1/15). Then ECE is computed as described in equation 5 (Appendix F).
>
> 8) Manuscript Changes
>
> We will make changes to the manuscript to reflect the reviewers other comments (typos/grammar), as well as include the suggested relevant works: https://arxiv.org/pdf/2003.03773.pdf, https://arxiv.org/abs/2006.07733, simCLR v2, https://arxiv.org/abs/2006.09882.

---

### Official Review · AnonReviewer4 · 2020-10-28
**Interesting approach with extensive experiments, some gaps remain**

**Rating:** 6
**Confidence:** 4

**Review:**

# Summary
This paper proposes uncertainty aware pseudo-labelling for semi-supervised learning, extending previously known methods by negative labels.

# Score justification
Well written paper and with extensive experiments with some points of improvement. The method should be positioned against others using confidence filtering and the role of calibration needs to be studied further.


# Strong and weak points
## Pros
Extensive experiments on different datasets and domains.
Broadly applicable method, that aims to improve conventional pseudo-labelling. Method is independent of uncertainty estimate and data augmentation.
Combining uncertainty estimates and confidence estimates, as well as adding negative learning are interesting approaches.

## Cons
Confidence filtering for pseudo-labels has already been suggested, see e.g. suggestions given in detailed comments.

I am unfortunately not convinced by the role of calibration, details given below. The authors could enhance the ablations studies with and without calibration and -importantly- adjusted thresholds, to clarify that point.



# Questions to the authors
The authors should consider e.g. https://arxiv.org/pdf/2002.02705 and https://www.microsoft.com/en-us/research/uploads/prod/2020/06/uncertainty_self_training_neurips_2020.pdf and position their work against those.

"Learning with UPS" have you treated pseudo-labels and original labels equally when training $f_{\theta,1}$, if so, how about fine-tuning on the original labels?

Please elaborate on the effect of different thresholds and how they were chosen. Especially with regards to calibration.

# Detailed comments
The authors should consider e.g. https://arxiv.org/pdf/2002.02705 and https://www.microsoft.com/en-us/research/uploads/prod/2020/06/uncertainty_self_training_neurips_2020.pdf

Augmentations for other domains are only not effective, if they are not suitable for that domain. Augmentations should use domain specific invariances. From the paper it becomes clear, you did use data augmentation, not sure why you argue so strongly against it.

I am not sure if calibration plays a role here. Calibration only moves the distribution in shape, by setting a confidence threshold $\tau$ the threshold would be changed, but a suitable threshold could be found before and after calibration.

Section 3.1 small type: "psuedo" --> "pseudo"

---

> ### Author Response · Authors · 2020-11-18
> **Response to Reviewer 4 (Augmentations, Fine-tuning, Calibration, and Thresholds)**
>
> We would like to thank the reviewer for their feedback and insightful comments. We appreciate the positive comments relating to the broad applicability of our method and our extensive experimentation. The following are responses to the reviewer's concerns.
>
>
>
> 1) Augmentations for other domains are only not effective, if they are not suitable for that domain... From the paper it becomes clear, you did use data augmentation, not sure why you argue so strongly against it.
>
> Many recent SSL works, like (Berthelot et al., 2020; Sohn et al., 2020), are designed with the assumption that a diverse set of domain-specific augmentations are available. UPS, on the other hand, is not designed with any specific augmentations in mind. Our aim is not to be against data augmentations (augmentations have been shown to work quite well for the SSL problem), but rather we attempt to show that competitive SSL performance can be achieved without inherently relying on such augmentations. We do use some augmentations during training, but we find that UPS still achieves better performance than other SSL approaches in the experimental setting where no augmentation is used (see response 1 to Reviewer3 for further details). This shows the broad applicability of UPS and that our method is not limited by the availability of a diverse set of augmentations.
>
>
> 2) have you treated pseudo-labels and original labels equally when training, if so, how about fine-tuning on the original labels?
>
> We treat the pseudo-labels and original labels equally when training our networks. Following the reviewer's suggestion, we further fine-tune the network with the original labels (for 50 epochs with a learning rate of 0.001) and analyse its effect on the pseudo-labeling accuracy. We observe that the average pseudo-labeling accuracy increases by 0.1\% to 0.4\% for CIFAR-10 with 4000 labels. This increase is consistent with the other CIFAR-10 and CIFAR-100 splits. This suggests that additional improvements can be obtained by incorporating this fine-tuning technique in UPS.
>
>
>
> 3) I am unfortunately not convinced by the role of calibration. The authors could enhance the ablations studies with and without calibration and -importantly- adjusted thresholds, to clarify that point.
>
>
> We show in our ablations (Table 5) that having a calibrated network with confidence-based thresholding achieves better performance than without calibration, when the confidence threshold are the same ($\tau_p$=0.7). However, we find that even with adjusted thresholds the calibrated network is unable to achieve sufficient pseudo-labeling accuracy to outperform the uncertainty-aware selection. We present the accuracy for the first set of selected pseudo-labels (CIFAR-100, 4000 labels) in the table below:
>
>
>  Threshold:                              |   t_p =0.7 | t_p =0.8 | t_p =0.9 |
>
> Conf.-Based Selection           | 54.92         | 58.75         | 64.18         |
>
> Conf.-Based Selection (Cal) | 64.75         | 70.48         | 77.18         |
>
> UPS                                           | 83.16         | 83.37         | 83.09         |
>
> Increasing the threshold for the calibrated network leads to increased accuracy (which is to be expected), but even with this high threshold of 0.9, it is unable to achieve the 83\% selected pseudo-label accuracy of UPS. Furthermore, we see that the addition of the uncertainty-aware selection allows UPS to be more robust to the confidence threshold, which allows it to be easily applied to different datasets.
>
> Although we believe that further improving network calibration would be a valid solution to this problem, which should warrant future study, the purpose of our work is not improving network calibration. Our calibration ablation is performed for completeness: since we obtain our uncertainty measure by performing MC-sampling, taking the mean of the different stochastic forward passes leads to more calibrated predictions.
>
>
> 4) Please elaborate on the effect of different thresholds and how they were chosen. Especially with regards to calibration.
>
> When we select our thresholds on the CIFAR-10 validation set (see Appendix H and response 4 to Reviewer2), we use the predictions which have been averaged over multiple stochastic forward passes. Therefore, these predictions are more calibrated than the standard network outputs. Since we select the uncertainty and confidence thresholds to obtain a sufficiently accurate subset of pseudo-labels, the network calibration only indirectly effects the hyper-parameter tuning.

---

> > ### Comment · AnonReviewer4 · 2020-11-25
> > **Comments on the changes**
> >
> >
> > I'd like to thank the authors for the additional insights provided.
> >
> > ## Comments
> > I'd like to emphasize the argument on calibration:
> > Using a single threshold for the calibrated and uncalibrated network is not meaningful. There should rather be a threshold $\vartheta_u$ for the uncalibrated and a threshold $\vartheta_c$ for the calibrated network. The temparature scaling method used in Guo et al., 2017, moves the distribution of confidences, it does not change it completely. After scaling the optimal threshold is lower (NNs being overconfident, scaling adjusts that), hence we should always expect $\vartheta_u > \vartheta_c$. If you would tune $\vartheta_u$ separately we can expect to achieve results on par with the  calibrated network. Simply the tuning is harder, since the separation between confidence values is more narrow as the distribution is "squeezed" in the high confidence regime. The method proposed by Guo et al, "expands" the distribution, such that the resulting confidences are more spread in the spectrum of possible values. This achieves better calibration, but for thresholding based approaches should make no fundamental difference.
> >
> > The experiments without data augmentation would be a suitable addition and should be combined with the other ablations studies.
> >
> > While I  appreciate the comment on fine-tuning I do not see it currently reflected in the paper.
> >
> > Small comment on the ablation study, please clarify the backbone used! WRN, S-S?
> >
> > ## Summary
> > The strong points raised in the original review remain, while I still see points of improvement as listed above. Hence I keep my original score.

---

> ### Author Response · Authors · 2020-11-18
> **Response to Reviewer 4 (Related Works)**
>
> 5) Confidence filtering for pseudo-labels has already been suggested, see e.g. suggestions given in detailed comments. The authors should consider e.g. https://arxiv.org/pdf/2002.02705 and https://www.microsoft.com/en-us/research/uploads/prod/2020/06/uncertainty_self_training_neurips_2020.pdf and position their work against those.
>
> Although confidence-based filtering has been used in previous works, our use of uncertainty-based selection is to account for the effect of poor network calibration in the pseudo-labeling process. The suggested papers are concurrent works and at the time of submission were not published.
>
> The first work (Haase-Schutz et al., 2020) partition an unlabeled dataset and train re-initialized networks on each partition. They use previously trained networks to filter the labels used for training newer networks. Most of their experiments involve learning from noisy data, and there is little focus on the SSL problem. The motivation for our work is to solve a fundamental issue with pseudo-labeling in SSL, by using an uncertainty-aware pseudo-label selection process that retains the simplicity and ease of implementation of conventional pseudo-labeling methods.
>
> The second work (Mukherjee et al., 2020) performs a similar learning strategy which selects pseudo-labels predicted by a pretrained language model using uncertainty for a downstream SSL task. One difference between our works is the selection of hard samples. Whereas Mukherjee et al. select a certain amount of hard samples (i.e. those which are not confident or certain) and learn from these using positive learning, we decide to use negative learning on these samples which reduces the amount of noise seen by the network.
>
> Our Related Works section will be updated with these two papers.

---

### Official Review · AnonReviewer2 · 2020-10-29
**The paper proposes a new pseudo-label selection method for pseudo-labeling based semi-supervised learning methods. The selection process labels positive and negative samples based on confidence and filter samples based on uncertainty. Experimental results show that the proposed method outperforms previous semi-supervised learning methods.**

**Rating:** 5
**Confidence:** 4

**Review:**

Strength:

The paper notices the problem in pseudo-labeling methods of semi-supervised learning, which is the erroneous prediction of pseudo-labeling. Pseudo-labeling is an easy-to-implement method for semi-supervised learning does not require constraints needed by consistency regularization methods, so improving pseudo-labeling methods can promote the practical use of semi-supervised learning.

The paper is well-written and easy to follow.

The experimental results on datasets of different domains such as image, video show that the proposed method outperforms previous semi-supervised learning methods.



Constraint:

The paper proposes a new method to predict pseudo-labels of unlabeled data by confident threshold. However, confidence threshold is a widely-used method to decide pseudo-labels. And the threshold of confidence is hard to decide since different backbone network and different datasets have different confidence levels.

Furthermore, using both positive and negative labels for classification is also used in (Kim et al., 2019). The authors fails to discuss the difference of the usage of positive and negative labels between their method and (Kim et al., 2019). The authors also need to discuss what is the special challenge of using positive and negative labels in pseudo-labeling for semi-supervised learning.

The paper argues that the calibration error is greatly reduced with more certain predictions. However, the paper only empirically shows the relation but fails to demonstrate the claims or give intuition on the relation. Also, confidence itself is also an uncertainty measure, but the authors do not use the confidence for uncertainty but use a new uncertainty measure. Could the authors explain what uncertainty measurement do they use and why using the new measurement instead of the confidence?

The paper uses a fixed set of threshold hyper-parameters for all the experiments. However, the confidence-level for different datasets should be different. For example, the confidence for a real difficult dataset should be much lower than an easy dataset. The authors need to show that why using a set of hyper-parameters is enough and how to select the hyper-parameters. For example, showing the relation between the accuracy of pseudo-labels and the confidence.

---

> ### Author Response · Authors · 2020-11-18
> **Response to Reviewer 2 (Confidence Threshold, Negative Learning, and Uncertainty Measurement)**
>
> We would like to thank the reviewer for their feedback and insightful comments. The following are responses to the reviewer's concerns.
>
>
> 1) "confidence threshold is a widely-used method to decide pseudo-labels. And the threshold of confidence is hard to decide..."
>
> We propose selecting pseudo-labels using both confidence and uncertainty thresholds. Whereas confidence thresholds have been used in previous works, the inclusion of uncertainty for this task is new. This uncertainty-aware thresholding allows our method to reduce the effect of poor network calibration in the pseudo-labeling process. We agree that different datasets may require different threshold hyper-parameters, but we show that our method performs reasonably well on multiple diverse datasets without performing dataset-specific tuning (see response 4 below for more information about hyper-parameter selection).
>
>
>
> 2) "The authors fails to discuss the difference of the usage of positive and negative labels between their method and (Kim et al., 2019)"
>
> We cite Kim et al. in the introduction since they are the first to propose the idea of negative learning (NL). There are several distinctions between the use of NL in the two works. First, the motivation of using negative labels in this work is to 1) incorporate more unlabeled samples into the training procedure, and 2) to generalize pseudo-labeling for the multi-label classification setting. On the other hand, Kim et al. use negative learning primarily to obtain good network initializations to learn with noisy labels.
>
> Furthermore, our negative labels are selected in an uncertainty-aware process (equation 5 in main paper), whereas Kim et al. initially generates negative labels randomly (NL step) to train a network and then use that network to selectively generate negative labels using confidence scores (SelNL). Their use of selective positive learning (SelPL) also relies on confidence-based positive pseudo-label creation. In our work we show that relying on confidence-based selection is insufficient, and our proposed uncertainty-aware selection is beneficial for the pseudo-labeling task. In general, our method is not built upon negative learning - we achieve strong performance without NL (see Table 5), but best performance is achieved when the additional negatively pseudo-labeled samples are used during training.
>
>
>
> 3) "Could the authors explain what uncertainty measurement do they use and why using the new measurement instead of the confidence?"
>
> Although confidence can be seen as a form of uncertainty, we find that it is insufficient to select pseudo-labels with predicted confidence scores because of the poor calibration of neural networks (Guo et al., 2017) - this is evident in our ablation (UPS, no UA) in Table 5, which shows that only using confidence-based selection (i.e. thresholding on predicted probability scores) achieves poor performance. For this reason, we incorporate an uncertainty measure that is not reliant upon specific probability outputs. To calculate this uncertainty measure, we use Monte Carlo sampling and obtain the uncertainty from the standard deviation of multiple stochastic forward passes (see section 4.1 and Appendix C). This uncertainty-aware selection procedure is a simple and easy to implement addition to the pseudo-labeling process, which leads to greatly improved pseudo-label accuracy. Furthermore, we show that our method can use several different uncertainty estimation methods while achieving comparable performance (MC-Dropout, MC-SpatialDropout, MC-DropBlock, and Data Augmentation), in Table 6.

---

> > ### Comment · AnonReviewer2 · 2020-11-18
> > **Post-rebuttal**
> >
> > Thanks the authors for the explanations on the paper. The idea of the paper is overall good but some places needs more clarification. I change my score to 5 since the paper is still not clarified well at submission, especially the discussion with the related works of positive and negative labels.

---

> > > ### Author Response · Authors · 2020-11-23
> > > **Revision Update**
> > >
> > > We thank the reviewer for their post-rebuttal feedback. We have revised the manuscript based on all reviewers' comments (see summary of changes in https://openreview.net/forum?id=-ODN6SbiUU&noteId=OSukB4ZBSk ). Notably, we have included further discussion and clarification about the use of negative labels in both the main text (page 2, footnote 1) and Appendix I.

---

> ### Author Response · Authors · 2020-11-18
> **Response to Reviewer 2 (Hyper-parameter Selection)**
>
> 4) "The authors need to show that why using a set of hyper-parameters is enough and how to select the hyper-parameters. For example, showing the relation between the accuracy of pseudo-labels and the confidence."
>
>
> We describe how hyper-parameters are selected in Appendix H. Our hyper-parameters are selected based on a 1000 sample CIFAR-10 validation set - the thresholds are selected such that a large amount of pseudo-labels would be selected while maintaining a relatively high accuracy. A graphical depiction of the independent relationships between pseudo-label accuracy and the confidence and uncertainty thresholds on this validation set can be seen in Figures 3a and 3b, respectively. On this validation set, we selected the confidence threshold of 0.7 and uncertainty threshold of 0.05 leading to the selection of 531 labels with 92.28\% accuracy. We find that once we select this uncertainty threshold (0.05), changes in the confidence threshold yields a similar numbers of selected pseudo-labels with similar accuracy. Although, if we select a less strict uncertainty threshold, then changes in the confidence threshold have larger impacts. Since we did not want to over-tweak the confidence threshold from dataset to dataset, we maintained this stricter uncertainty threshold of 0.05 throughout our experiments. However, for any drastically different dataset, we can perform this analysis to obtain a new set of thresholds.
>
> Using a fixed set of hyper-parameters in our experiments demonstrates that UPS can give reasonable performance without dataset specific hyper-parameter tuning; we achieve strong performance on CIFAR-100 and UCF-101 with thresholds obtained from the CIFAR-10 validation set. It is true that a better set of hyper-parameters for a particular dataset can always be found, which is also the case for existing SSL methods (e.g. loss weighting for unlabeled samples and $\alpha$ in the MixUp augmentation are both hyper-parameters which can be tuned for improved performance), but the robustness of our method (shown on multiple datasets) and a reasonable hyper-parameter selection strategy enables our method to be applicable to many different datasets.

---

### Official Review · AnonReviewer3 · 2020-10-31
**Through taking into account miscalibration in modern neural networks, this works produces a competitive Semi-Supervised Learning approach based Pseudo-Labeling**

**Rating:** 6
**Confidence:** 4

**Review:**

As noted in (Guo et al., 2017), modern neural networks are often miscalibrated. Pseudo-labeling based Semi-Supervised learning schemes are predicated on high confidence predictions from these neural networks. This paper posits that this miscalibration may lead to inferior results in confidence-based pseudo-labeling approaches. By taking into account the uncertainty of models and only using pseudo labels from high-confidence instances with low uncertainty this work presents a model that significant improves on other PL strategies and is competitive with consistency-based regularization strategies that comprise the current state-of-the-art.

Not only does this approach generate competitive results from with pseudo labeling strategy (which is compelling due to the history of PL approaches), but it is also less reliant on domain-specific augmentations that current consistency-based regularization approaches rely on. This is important for extending approach beyond domains where specific types of augmentation have been extensively studied. The results on the video domain offer some evidence of the usefulness in under-explored domains.


For the CIFAR (10/100) results presented it is unclear what data augmentations where used to produce the UPS results and how important these augmentations are to presented performance. As a key claim in this paper is de-emphasis of domain-specific augmentations would be beneficial to see a highlighted result strengthening this claim.

Also, it seems the model is re-initialized and trained to convergence after each round of pseudo labeling. This would extremely compute intensive for large-scale semi-supervised learning tasks, which is a motivating use-case for semi-supervised learning. Providing results and comparisons on the compute requirements for this approach and its comparisons with competitors would be beneficial. As the re-initialization, is likely a major component in compute resources would also be beneficial for community to understand how essential this is to the improved performance numbers presented in the paper.

---

> ### Author Response · Authors · 2020-11-18
> **Response to Reviewer 3  (Data Augmentation and Compute Requirements)**
>
> We would like to thank the reviewer for their feedback and insightful comments. We appreciate the positive comments relating to UPS's competitive performance as well as its extensions to other domains, specifically video. The following are responses to the reviewer's concerns.
>
> 1) "what data augmentations where used to produce the UPS results and how important these augmentations are to presented performance."
>
> During training on image datasets, we use RandAugment; for experiments on UCF-101, we use random crop and temporal jittering (see Appendix D). Following the reviewer's suggestion, we have run additional experiments on CIFAR-10, with no input augmentations. The results are shown in the table below. Our method achieves an error rate of 28.14\% and 14.98\% for 1000 and 4000 labels respectively. This is a respectable score that improves upon other SSL methods Π model (Samuli Laine and Timo Aila, 2017) and Mean Teacher (Tarvainen & Valpola, 2017) in the same experimental setting (i.e. no data augmentations).
>
> Method                     | 1000 labels  | 4000 labels
>
> Π model[1]*             | 32.18            | 17.08
>
> Mean Teacher[2]*   | 30.62            | 17.74
>
> UPS                             | 28.14            | 14.98
>
> *The scores are obtained from Mean Teacher[2].
>
> Another set of experiments which illustrates that our method is not inherently reliant on domain-specific augmentations are those on UCF-101. In the video domain, there are a limited number of augmentations during training, and UPS is able to achieve strong performance in this setting. These experiments, along with our experiments in Appendix D, show that UPS is not reliant on any specific set of data augmentations, but it can take advantage of available augmentations to further improve performance.
>
>
> 2) "The compute requirements for this approach"
>
> For the more complex datasets (Pascal VOC and UCF-101), we find that UPS requires relatively few pseudo-labeling iterations (re-initializations). UCF-101 requires at most 2 re-initializations, while Pascal VOC requires at most 4. We agree that this incurs a larger computing overhead than previous SSL approaches, but the network re-initialization limits error propagation from previous training and pseudo-labeling steps. Making this training scheme more efficient, would be an interesting extension to this work.
>
> [1] Samuli Laine and Timo Aila. Temporal Ensembling for Semi-Supervised Learning. International Conference on Learning Representations, {ICLR} 2017.
>
> [2] Antti Tarvainen and Harri Valpola. Mean teachers are better role models: Weight-averaged consistency targets improve semi-supervised deep learning results. In Advances in Neural Information Processing Systems, 2017.

---

### Author Response · Authors · 2020-11-23
**Summary of Initial Revision**

Following the reviewers' feedback, we have updated the manuscript with the following changes:

* Extended the related works by incorporating the relevant related works.
* Updated the section D of Appendix to include the experiment without any input augmentation.
* Updated the section H of Appendix with additional details about hyperparameter selection.
* Added clarification about the use of negative learning in page 2, footnote 1 and section I of appendix.
* For further clarification we have mentioned $\tau_p$ $\ge$ $\tau_n$ after equation 2.
* Updated the caption for figure 1 to specify on which dataset the analysis has been performed to generate figure 1(b) and 1(c)
* Updated the caption of Table 1 to clarify which methods are PL based and which are consistency regularization based.
* Updated the caption of figure 2 to specify on which dataset the experiment has been performed.
* Fixed the issues with spelling and punctuation.

Once again, we thank the Reviewers and Area Chair for their time and helpful comments.

---

### Decision · Program_Chairs · 2021-01-07
**Final Decision**

**Decision:**

Accept (Poster)

**Comment:**

The paper is written in defense of pseudo-labeling. The authors `aim at demonstrating that pseudo-labeling based methods can perform on par with consistency regularization methods which have been show to achieve strong performance.

The paper is well-written and easy-to-follow. The reviewers are generally positive about the contribution. It has to be underline, however, that pseudo-labeling is still a controversial approach with a very limited theoretical understanding. This paper does not provide any further understanding of it, but proposes several "heuristics", intuitively well-motivated, but justified only experimentally. Nevertheless, the results are promising and the paper is an important voice in the general discussion around learning with weak labels and semi-supervised learning, two crucial problems in many practical applications. Taking this into account I recommend to accept the paper as a poster.

The reviewers have raised several problems that the authors have been exhaustively discussing in their rebuttals. The one remaining issue is the interaction between calibration and the threshold. This problem has to be clarified in the final version of the paper, as indeed calibration usually does not change the order.